# Sustainability of the Slovak Spirits Industry in the Single Market of the EU

Ondrej Beňuš *, Peter Bielik, Natália Turčeková and Izabela Adamičková

Department of Economics, Faculty of Economics and Management, Slovak University of Agriculture in Nitra, 949 76 Nitra, Slovakia; Peter.bielik@uniag.sk (P.B.); natalia.turcekova@uniag.sk (N.T.); izabela.adamickova@uniag.sk (I.A.)
* Correspondence: ondrej.benus@uniag.sk

**Abstract:** Despite a decreased share of gross domestic product, the role of a country's food and beverage industry cannot be underestimated. Food security should be a crucial part of national policy in every country because, during critical situations, such as the COVID-19 pandemic, it plays a key role in the protection of a country's citizens. The food industry also plays a key role in employment and the sustainable development of all countries. We investigated the latest trends and focused on one specific branch of the food industry. Although the spirits industry does not have a major market share in the food industry, it is one of the branches with the highest added value, and it produces commodities with high export potential. This branch of the food industry in the Slovak Republic has a long tradition. Our primary aim was to examine the competitiveness of this sector in a single market of the EU as a key element of its sustainable development. The results obtained for the revealed comparative advantage suggest that there was a downturn in the number of competitive branches in the Slovak spirits industry during the study period of 2004–2018. Despite this negative trend, the remaining competitive branches of the Slovak spirits industry represent the majority of exports.

**Keywords:** spirits industry; single market of the EU; competitiveness; revealed comparative advantage





## 1. Introduction

The food and drink industry of the European Union represents the largest manufacturing sector [1]; thus, its performance and competitiveness are among the key drivers of national economies. The agri-food sector and, especially, agricultural commodities have stagnated for the last few decades. In developed countries in particular, both agriculture and the food industry are losing their market share to innovative and more technologically based industries. However, despite this trend, the agri-food industry is a key industry in every national economy that is responsible for food safety. The competitiveness of this sector has been of central interest to policymakers, entrepreneurs, and researchers. This sector needs to be preserved in order to ensure long-lasting and stable employment. New industries generate faster growth in national economies but are usually sensitive to economic downturns. The COVID-19 pandemic has shown that traditional branches of industries, with the food industry at the forefront, have been able to better withstand economic downturn pressures compared to many rising industries over the last few decades [2,3]. This resilience is very important, as the agri-food industry is still a key employer in national economies. The food and drink industry employs 4.72 million people in the EU [1]: it is a major employer in 15 EU countries and the second most important employer in a further seven EU countries [1]. In our view, investigating the position of the agri-food sector is important for obtaining information on its sustainable development. The latest research shows that the sustainable development of the agri-food sector is correlated with the international competitiveness of selected commodities [4]. This specialization may help preserve the sustainability of the agri-food sector as a whole.

The available literature shows that a significant amount of research has been dedicated to this field in recent years [5–9]. Within Europe, there has been substantial research on agri-food competitiveness [10,11]. Competitiveness investigations have been performed at various levels, ranging from cross-European to regional entities.

From a review of research activities, we can identify three independent areas of study:

- Competitiveness of the EU (or other regional collaborative formats) at the national level [6,8];
- Competitiveness of single countries [9,10]; and
- Competitiveness of specific agri-food products [7,11].

Studies in the first area were oriented toward the competitiveness of agri-food in the entire EU. The latest competitiveness research on EU agriculture has been dedicated to the impacts of economic crisis, which played a key role in the development of competitiveness within this sector [12].

However, research became even more frequent with the addition of new member states to this portfolio in 2004 [13,14].

The authors started to investigate the true impacts of EU membership on these countries. Agri-food trade was considered equal between old member states (EU15) and new member states (EU12 after 2004), but the number of quality products with a higher added value was greater in old member states (EU15) [14,15].

Other researchers [5,9] found that an increasing export tendency of agri-food products was apparent within new member states after they joined the EU; however, the import share was still higher and grew continually. This became a growing concern when the dependence on a single market became evident. Export was identified in a similar way to previous research outcomes when the orientation of raw agri-food material exports of the new member states became dominant. Imported products were represented by processed products (based on later research outcomes) [6].

Investigating the impacts of EU accession, Kiss [15] added new evidence on the agri-food trade of new member states in a single market. The analysis of data between 2000 and 2010 revealed the growth of competition in the domestic markets of new member states. Reductions in employment and the growth of salaries in the agri-food sector were also apparent.

The authors also identified significant differences in competitiveness in agri-food trade within new member states. Svatoš and Smutka [16] investigated the competitiveness of the agri-food sector among the Central European countries. The export value grew in all of the Central European countries, but the overall competitiveness of agri-food trade differed between them, according to the authors. The Czech Republic and the Slovak Republic observed a decrease in competitiveness after joining the EU. On the other hand, Hungary and Poland experienced an increase in competitiveness in their agri-food sectors. This evidence was also supported by additional researchers, who highlighted the different positions of Central European countries [17]. The decrease in the competitiveness of agri-food products was also confirmed by further research within the Slovak Republic and the Czech Republic, as a concentration of products with a minimum level of added value was observed [18]. This phenomenon was further investigated by other authors, and there is some new evidence available. Harvey et al. [6] considered productivity within the agri-food sector to be one of the most important drivers of further development. They found a significant difference between new member states and old member states. While old member states have experienced positive growth in productivity, new member states have struggled to maintain the same level of productivity [19]. As a result, new national strategies for the agri-food industry have emerged [20].

Previous research on agri-food competitiveness in Southeast Europe has also shown significant differences between regionally close countries and highlighted the importance of individual support systems in different branches of the agri-food sector [10,21]. The specialization of value-added products shows that this step is related to an improvement in the competitiveness of the highly competitive single market of the EU [22].

The latest research tends to be more oriented toward particular branches of the agri-food sector [23]. This approach helps to precisely identify branches that represent competitive export commodities within investigated countries. The main research activity has been concentrated in the dominant branches of the agri-food industry. The milk industry is one of the key agri-food branches, and it represents a common research focus of competitiveness. In this segment, competitiveness is significantly disproportionate among European countries [24]. When investigating individual commodities within the dairy industry, many authors established further disparity within the same country. Dusan et al. [25] investigated the competitiveness of the Slovak dairy industry. His calculations clearly demonstrate different trends within this branch of the Slovak food industry, and only one commodity was competitive in the international market. There is also some research on the production of alcoholic beverages within the EU. The wine industry is the most researched alcoholic beverage branch among studies on competitiveness. An investigation of the wine industry has shown clear evidence that traditional wine-producing countries in the EU have experienced a decrease in competitiveness over the past several years [26]. These countries are facing rising competition from countries outside of the EU, and evidence suggests that only a small number of European countries can preserve their competitive advantage in wine production over the long term [27]. Other research has focused on the beer market and its competitiveness. Research examining the industry at a global scale has shown that export competitiveness is strongly concentrated within only a few traditional beer-producing countries in the world [28]. We also identified research oriented toward the competitiveness of small beer producers that represent the craft beer movement [29]. To date, research focused on the spirits industry has been scarce. Although findings have been reported, they mostly focus on particular products and not the industry as a whole. For example, research on fruit spirits, a traditional product in Central and Eastern European countries, showed a decline in competitiveness between 2001 and 2011 [30].

To follow up on the latest research, we investigated the development of the Slovak spirits industry in terms of its competitiveness within a single market of the EU.

## 2. Materials and Methods

The main aim of this study was to investigate the competitiveness of the Slovak spirits industry after the country joined the European Union. We describe seven specific branches of the Slovak spirits industry in the following section. The spirits industry has been identified as a "competitive" branch by the Ministry of Agriculture of the Slovak Republic in its "Concept of development of the Food Industry 2014–2020" [20]. As we near the end of the projected time period, we reason that now is the right time to measure the initial outcomes of the measurements established by policymakers.

### 2.1. Research Questions

To fulfill our primary aim, our research was guided by two research questions:

1. How has the number of competitive branches of the Slovak spirits industry changed on a single market of the EU during the observed years?

We investigated the change in competitiveness for each branch of the Slovak spirits industry. The aim is determine whether there is overall growth or decline in competitiveness across all branches of the Slovak spirits industry. The research outcomes also reveal the inner dynamics of individual branches of the Slovak spirits industry during the observed years.

2. How has the total export share of competitive branches of the Slovak spirits industry changed during the observed years in a single market of the EU?

The answer to this research question determines the overall competitiveness of the Slovak spirits industry on a single market of the EU. A positive change in this area is crucial and even more important than the total number of competitive branches of the

spirits industry. In particular, growth in total share would indicate the specialization of production in a single market of the EU.

*2.2. Research Focus*

We focused our research on individual branches of the spirits industry in the Slovak Republic and their export/import development throughout the observed years. We used well-established methods for measuring competitiveness within industries and focused on one particular industry within the Slovak Republic that has not been previously investigated.

For our calculations, we classified product groups using the Combined Nomenclature based on the Council Regulation (EEC) No. 2658/87 of 23 July 1987 on the tariff and statistical nomenclature and on the Common Customs Tariff.

We investigated products within chapter 22 of the Combined Nomenclature. This chapter has two basic commodity codes dedicated to ethyl alcohol:

- 2207—Undenatured ethyl alcohol of an alcoholic strength by volume of 80% vol. or higher; ethyl alcohol and other spirits, denatured, of any strength;
- 2208—Undenatured ethyl alcohol of an alcoholic strength by a volume of less than 80% vol.; spirits, liqueurs, and other spirituous beverages (see Table 1).

**Table 1.** Nomenclature codes within subsection 22 of chapter 22 according to the Council Regulation (EEC) No. 2658/87 of 23 July 1987 on the tariff and statistical nomenclature and on the Common Custom Tariff.

| Nomenclature Code | Definition |
|---|---|
| 220820 | Spirits obtained by distilling grape wine or grape marc |
| 220830 | Whiskies |
| 220840 | Rum and other spirits obtained by distilling fermented sugarcane products |
| 220850 | Gin and Geneva |
| 220860 | Vodka |
| 220870 | Liqueurs and cordials |
| 220890 | Ethyl alcohol of an alcoholic strength of <80% vol, not denatured; spirits and other spirituous beverages |

As revealed in the definitions of the two abovementioned categories, the main difference is the strength of ethyl alcohol. The 2207 category contains only spirits with an ethyl alcohol percentage above 80%. This group of spirits is not precisely divided and generally contains ethyl alcohol products for further mixing or applying other processing methods. Our main aim was to investigate final products within the spirits industry. Therefore, we only investigated final products listed under subsection 2208 of chapter 22 of the Combined Nomenclature.

The abovementioned nomenclature is explained in Regulation (EC) No. 110/2008 of the European Parliament and of the Council of 15 January 2008 on the definition, description, presentation, labelling, and protection of geographical indications of spirit drinks and repealing Council Regulation (EEC) No. 1576/89.

Code 220820 (spirits obtained by distilling grape wine or grape marc) addresses wine spirits and grape marc. Wine spirits are drinks produced exclusively by distillation at less than 86% vol. of wine or wine fortified for distillation or by the redistillation of a wine distillate at less than 86% vol. Grape marc is defined by the abovementioned regulation as a spirit produced exclusively from grape marc fermented and distilled either directly by water vapor or after water has been added.

Whiskies are listed under code 220830. Their general description refers to a spirit distillation of a mash made from malted cereals with or without the whole grains of other cereals that have been

- Saccharified by the diastase of the malt contained therein, with or without other natural enzymes, or

- Fermented by the action of yeast.

Code 220840 contains rum and other spirits obtained by distilling fermented sugarcane products. These products represent spirits produced exclusively by alcoholic fermentation and distillation, either from molasses, syrup produced in the manufacture of cane sugar, sugarcane juice, or sugarcane juice having aromatic characteristics specific to rum.

Gin and Geneva, as listed under code 220850, represent a broad range of spirits made by using juniper in various forms. Gin is the main product in this group. It is a juniper-flavored spirit drink produced by flavoring organoleptically suitable ethyl alcohol of agricultural origin with juniper berries.

Code 220860 sets the legal definition of vodka products. Vodka is among the most actively traded products within the spirits industry around the world. According to the cited regulation, vodka is a spirit drink produced from ethyl alcohol of agricultural origin obtained following fermentation with yeast from either potatoes, cereals, or other agricultural raw materials.

Code 220870 contains liqueurs and cordials. The regulation defines a liqueur as a spirit produced by flavoring ethyl alcohol of agricultural origin; is a distillate of agricultural origin; contains one or more spirit drinks or a mixture thereof; and is sweetened with the addition of products of agricultural origin or foodstuffs such as cream, milk or other milk products, fruit, or wine or aromatized wine.

The last investigated category is code 220890, which represents the most diverse group of products. There are products listed that do not fall into any of the previous categories and represent spirits (under 80% of ethyl alcohol) for human consumption. Among the most important spirits within this category are the following:

- Arrack;
- Plum, pear, or cherry spirit;
- Ouzo;
- Calvados; and
- Tequila.

*2.3. Research Methodology*

The competitiveness of the Slovak spirits industry was measured within the territory of the European Union. We also included the United Kingdom because it was part of the EU during the investigated period of time. This approach enabled us to focus our research exclusively on intra-EU trade as defined by Regulation (EC) No. 638/2004 of the European Parliament and the Council of 31 March 2004 on community statistics relating to the trading of goods between the member states and repealing Council Regulation (EEC) No. 3330/91. The measurements reported in this manuscript represent the competitiveness of the spirits industry in a single market of the EU and can unveil a change in competitiveness in this industry. However, by excluding the extra-EU trade of the Slovak spirits industry, we also risked the loss of information. The question was the relevance of the extra-EU trade to total exports. If the volume of agri-food trade outside of the EU was high, then the conclusions of the proposed research would potentially be misleading. Previous research has examined [31] differences in agri-food trade between developed and less developed countries in the international market. Agri-food production in developed countries is focused on intranational trade. Agricultural products in developed countries are processed by enterprises situated within the national economy, and most of the food production is dedicated to local (national) consumption. The exports of developed countries tend to be goods and services with higher added value. However, less developed countries are typically engaged in international trade of agri-food products. This is usually due to a manufacturing sector that is significantly less developed compared to developed countries. Lacking processing capacities, less developed countries focus on the export of agricultural products. These agricultural products are often further processed in developed countries in order to justify national consumption. Considering the economic development level of

Central European countries, we infer that the majority of agri-food products are sold on a single market of the EU than outside of the EU.

Data were collected for the years 2004–2018. The starting point of our research period was the year when the Slovak Republic joined the European Union. The year 2018 was the latest available dataset that could be obtained at the time of the research. The source for our calculations was a dataset (HS2-HS4) provided by the Eurostat database on international trade. These data include the value and quantity of goods traded between EU member states and those traded by EU member states with non-EU countries. Our calculations used the values of trade within the EU. The research methodology is described in Figure 1.

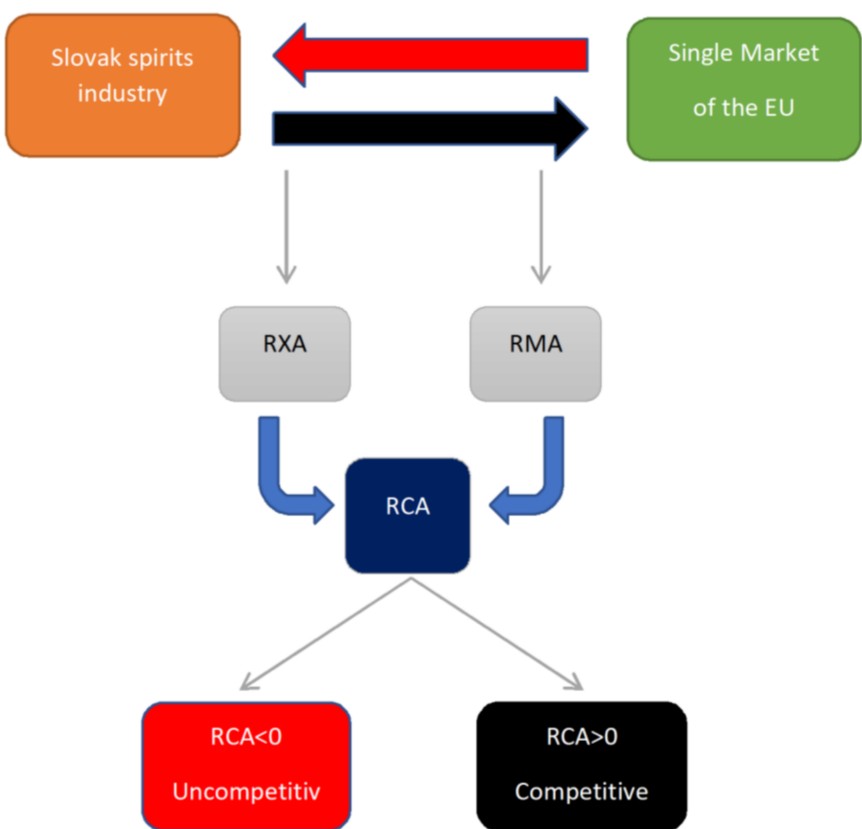

**Figure 1.** Research methodology for measuring competitiveness in the Slovak spirits industry.

Based on our measurements, we divided all observed product groups, as defined by the nomenclature, into two separate groups based on the revealed comparative advantage (RCA) with the following criteria:

- Revealed competitiveness advantage > 0;
- Revealed competitiveness advantage < 0.

The RCA value used to define the current competitiveness level was the value identified for each code in 2018. We applied this rule in order to show the competitiveness of each product group according to the latest data indicated by the Eurostat datasets.

The first group of products is represented by competitive branches of the Slovak spirits industry in the single market (RCA > 0). These products are crucial for determining the landscape of competitiveness in the Slovak spirits industry. The second group (RCA < 0) represents those products that did not achieve a competitive position in the single market of the EU. The share of total exports in both groups mentioned determines whether the majority of exports hold a competitive position within the European market.

*2.4. Research Methods*

We used the revealed comparative advantage as the main indicator for measuring the competitiveness of the Slovak spirits industry. This approach has been widely applied in previously published research [4,9,10,12,17] that is in line with our research objectives.

The main measurement method was the calculation of competitiveness as first introduced by Balassa [32], who was a pioneer in competitiveness analysis. This approach is widely used by researchers in various fields. Balassa [32] measured comparative advantage by using only export data. Balassa's [32] competitiveness advantage can be measured according to the following formula:

$$RCA_a^i = \left(\frac{X_a^i}{X_a^c}\right) \Big/ \left(\frac{X_m^i}{X_m^c}\right) \tag{1}$$

where *X* represents the export, the subscript *m* refers to combined exports, *A* refers to a particular product, *i* refers to a particular country, and *c* refers to all observed countries.

Despite the broad use of the abovementioned formula, we expanded this equation using a wide variety of modified approaches to measuring competitiveness [33–35]. We used the approach first introduced by Vollrath [34], which includes import value in the measurement of comparative advantage. According to the measurements used by Vollrath [34], our research is based on three different indicators:

RXA—relative export advantage:

$$RXA_a^i = \left(\frac{X_a^i}{X_n^i}\right) \Big/ \left(\frac{X_a^r}{X_n^r}\right) \tag{2}$$

RMA—relative import advantage:

$$RMA_a^i = \left(\frac{M_a^i}{M_a^c}\right) \Big/ \left(\frac{M_m^i}{M_m^c}\right) \tag{3}$$

RTA—relative trade advantage:

$$RTA_a^i = RXA_a^i - RMA_a^i \tag{4}$$

RCA—revealed competitiveness:

$$RCA_a^i = Ln(RXA_a^i) - Ln(RMA_a^i) \tag{5}$$

where *X* represents export, *M* refers to import, *i* refers to a selected country, *r* refers to all countries (except country i), *a* refers to a specific product, and *n* refers to all products (except product *a*).

All three indicators measured for the 14 observed years were also used for the descriptive statistics of each product group.

## 3. Results

When comparing the share of each branch within the Slovak spirits industry, we can clearly identify significant changes between the observed years. In the first year of our research period, the three most important export commodities were registered under the following nomenclature codes:

- 220890 (ethyl alcohol of an alcoholic strength of <80% vol, not denatured; spirits and other spirituous beverages);
- 220870 (liqueurs and cordials);
- 220820 (spirits obtained by distilling grape wine or grape marc).

Together, these three product groups accounted for 77.4% of the spirits exported in the first observed year. If we compare these values with those in 2018, significant changes are evident. Dynamic changes are observed in the export of the three most exported commodities in the Slovak spirits industry (see Table 2). Products under code 220820 (spirits obtained by distilling grape wine or grape marc), previously the third most exported commodity, experienced a decline of 16% within the observed period of time. Similarly, products under code 220890 (ethyl alcohol of an alcoholic strength of <80% vol, not denatured; spirits and other spirituous beverages) experienced a decrease of 14.7%. On the other hand, the product group under code 220870 (liqueurs and cordials) was able to improve its export share by 16.6% between 2004 and 2018. This change caused this branch of the Slovak spirits industry to become the most important export commodity in the final year, 2018. The results of our research are strongly influenced by the competitiveness of this commodity, as its 43.8% share in the total export value shaped the competitiveness of the whole Slovak spirits industry in last year.

**Table 2.** Share (%) of total spirit exports for each investigated commodity between 2004 and 2018.

|  | 2004 | 2005 | 2006 | 2007 | 2008 | 2009 | 2010 | 2011 | 2012 | 2013 | 2014 | 2015 | 2016 | 2017 | 2018 |
|---|---|---|---|---|---|---|---|---|---|---|---|---|---|---|---|
| 220820 | 19.8 | 8.3 | 3.7 | 2.9 | 0.3 | 0.1 | 0.3 | 0.4 | 0.2 | 1.4 | 0.4 | 1.1 | 0.6 | 0.9 | 3.8 |
| 220830 | 0.1 | 1.3 | 2.9 | 0.0 | 0.7 | 1.4 | 0.4 | 3.2 | 0.0 | 1.0 | 27.3 | 4.1 | 13.6 | 20.1 | 14.5 |
| 220840 | : | : | 1.3 | : | 0.0 | 0.0 | 0.1 | 0.2 | 0.2 | 0.1 | 1.0 | 3.8 | 4.4 | 7.1 | 6.5 |
| 220850 | 4.3 | 3.4 | 3.5 | 3.5 | 8.4 | 4.0 | 1.4 | 1.5 | 1.4 | 0.7 | 0.7 | 1.7 | 2.5 | 2.4 | 2.4 |
| 220860 | 18.2 | 23.4 | 30.5 | 31.0 | 15.0 | 5.4 | 7.7 | 10.7 | 16.2 | 18.7 | 15.8 | 20.3 | 16.0 | 15.9 | 13.4 |
| 220870 | 27.2 | 34.9 | 21.0 | 24.2 | 34.1 | 55.8 | 55.9 | 42.2 | 35.0 | 24.5 | 25.4 | 35.8 | 36.4 | 29.3 | 43.8 |
| 220890 | 30.4 | 28.7 | 37.1 | 38.4 | 41.5 | 33.3 | 34.2 | 41.8 | 46.9 | 53.6 | 29.4 | 33.2 | 26.5 | 24.4 | 15.7 |

We compared the growth potential of the observed branches of the Slovak spirits industry (with emphasis on export values) during the observed period of time, and dynamic growth was observed in some branches. Among the new key players in the industry are products registered under code 220830 (Whiskies). Recently, these products became the third most imported commodity within the spirits industry in the Slovak Republic (see Table 3). However, the growth of the import value of this commodity was significantly lower compared to the export growth measured within the same period of time. Another new product category worth closer investigation is rum and its co-products registered under code 220840 (rum and other spirits obtained by distilling fermented sugarcane products). This product group experienced significant growth in both export and import values during the observed period of time.

**Table 3.** Share (%) of total spirit imports for each investigated commodity between 2004 and 2018.

|  | 2004 | 2005 | 2006 | 2007 | 2008 | 2009 | 2010 | 2011 | 2012 | 2013 | 2014 | 2015 | 2016 | 2017 | 2018 |
|---|---|---|---|---|---|---|---|---|---|---|---|---|---|---|---|
| 220820 | 14.8 | 14.5 | 19.0 | 19.4 | 14.0 | 21.8 | 12.8 | 11.5 | 11.6 | 12.8 | 13.0 | 14.2 | 17.9 | 14.9 | 9.8 |
| 220830 | 14.1 | 12.5 | 13.1 | 13.7 | 13.5 | 9.5 | 12.6 | 14.4 | 17.3 | 16.2 | 18.7 | 16.3 | 16.1 | 19.4 | 18.1 |
| 220840 | 1.0 | 1.2 | 2.2 | 2.7 | 2.7 | 2.3 | 2.8 | 3.6 | 4.7 | 8.3 | 9.9 | 12.0 | 13.3 | 13.6 | 15.1 |
| 220850 | 2.7 | 2.4 | 2.1 | 1.6 | 2.0 | 2.1 | 2.9 | 2.7 | 2.5 | 3.0 | 3.5 | 3.4 | 3.5 | 4.2 | 4.3 |
| 220860 | 7.9 | 7.0 | 8.9 | 9.1 | 10.7 | 10.8 | 11.1 | 12.3 | 15.1 | 15.0 | 14.2 | 13.8 | 12.6 | 12.0 | 12.5 |
| 220870 | 27.1 | 29.3 | 18.5 | 19.0 | 21.3 | 17.2 | 18.0 | 16.3 | 16.1 | 15.4 | 15.2 | 16.4 | 18.8 | 13.5 | 16.8 |
| 220890 | 32.4 | 33.1 | 36.3 | 34.5 | 35.9 | 36.3 | 39.8 | 39.3 | 32.7 | 29.2 | 25.6 | 24.0 | 17.9 | 22.4 | 23.4 |

The investigation of import values in 2004 revealed three dominant import commodities registered under the following codes:

- Code 220890 (ethyl alcohol of an alcoholic strength of <80% vol, not denatured; spirits and other spirituous beverages);
- Code 220870 (liqueurs and cordials);
- Code 220820 (spirits obtained by distilling grape wine or grape marc).

If we compare export and import data, we can clearly see similarities within the investigated commodities. In 2004, the three most important branches of the spirits industry in terms of international trade were the same between export and import data. The import declines in the branches mentioned (220820, 220870, and 220890) were lower compared to the dynamics observed in exports. We measured the largest import decline in products under code 220870 (liqueurs and cordials). This outcome differs from the trend in export values of this commodity, which experienced the highest growth of export value.

From the available export and import structure data between 2004 and 2018, we can already recognize that the Slovak spirits industry has adapted to the EU market. Based on our initial calculations of the Slovak spirits export and import structure, we decided to conduct a more in-depth analysis of the competitiveness of the Slovak spirits industry in a single market of the EU.

First, we calculated the descriptive statistics of the relative export advantage, relative trade advantage, and revealed comparative advantage within all investigated branches. We computed descriptive statistics in two separate groups. The first group (as shown in Table 4) refers to all product categories that experienced a negative mean RCA value during the observed years on average. On the other hand, the second group (as shown in Table 5) refers to "competitive" commodities of the Slovak spirits industry. Product groups with positive mean values were those registered under codes 220860, 220870, and 220890. All of these product groups also had the lowest standard deviations among all branches of the Slovak spirits industry. From these data, the variation coefficients of all three product groups mentioned were positive in contrast to all other investigated codes. The highest positive mean value was observed within code 220870 (liqueurs and cordials). Products in this group experienced a steep increase in export share and represented the most significant commodity within Slovak spirit exports. On the other hand, the share of import value of this commodity has declined by 9%. Code 220890 had the lowest RCA variance and standard deviation between 2004 and 2018. The mean value of the RCA standard deviation was highest for code 220860 because its export value underwent the largest change over the last 15 years among all observed product groups. In fact, we measured the highest export share of 10.7% in 2004 and the highest share of 31% in 2007.

**Table 4.** Descriptive statistics of measured indicators with mean RCA values below 0 for each observed code within 2004–2018.

| | 220820 | | | 220830 | | | 220840 | | | 220850 | | |
| --- | --- | --- | --- | --- | --- | --- | --- | --- | --- | --- | --- | --- |
| | **RXA** | **RTA** | **RCA** | **RXA** | **RTA** | **RCA** | **RXA** | **RTA** | **RCA** | **RXA** | **RTA** | **RCA** |
| Nbr. of observations | 15.00 | 15.00 | 15.00 | 15.00 | 15.00 | 15.00 | 15.00 | 15.00 | 15.00 | 15.00 | 15.00 | 15.00 |
| Nbr. of missing values | 0.00 | 0.00 | 0.00 | 1.00 | 0.00 | 1.00 | 3.00 | 3.00 | 3.00 | 0.00 | 0.00 | 0.00 |
| Minimum | −4.82 | −2.02 | −5.53 | −8.30 | −0.35 | −7.26 | −5.20 | −1.56 | −4.32 | −2.15 | −0.60 | −1.65 |
| Maximum | 0.57 | 0.43 | 0.28 | −0.55 | 0.23 | 0.50 | 0.03 | −0.25 | −0.66 | 0.87 | 1.69 | 1.35 |
| First quartile | −3.61 | −1.42 | −3.80 | −4.43 | −0.25 | −3.15 | −3.66 | −1.43 | −3.25 | −1.24 | −0.36 | −0.78 |
| Median | −2.58 | −1.05 | −3.07 | −3.50 | −0.22 | −1.93 | −2.46 | −0.90 | −2.59 | −1.09 | −0.21 | −0.54 |
| Third quartile | −1.66 | −0.83 | −2.04 | −1.70 | −0.12 | −0.57 | −0.51 | −0.54 | −1.09 | −0.06 | 0.46 | 0.69 |
| Mean | −2.57 | −1.08 | −2.81 | −3.53 | −0.17 | −2.25 | −2.32 | −0.96 | −2.36 | −0.78 | 0.07 | −0.18 |
| Variance (*n*) | 2.14 | 0.32 | 2.10 | 4.49 | 0.02 | 4.25 | 3.15 | 0.23 | 1.74 | 0.68 | 0.38 | 0.95 |
| Standard deviation (*n*) | 1.46 | 0.57 | 1.45 | 2.12 | 0.14 | 2.06 | 1.77 | 0.48 | 1.32 | 0.82 | 0.61 | 0.98 |
| Variation coefficient (*n*) | −0.57 | −0.53 | −0.51 | −0.60 | −0.86 | −0.91 | −0.76 | −0.51 | −0.56 | −1.06 | 8.64 | −5.51 |
| Skewness (Pearson) | 0.48 | 0.71 | 0.39 | −0.54 | 1.37 | −0.82 | −0.07 | 0.01 | −0.10 | 0.17 | 1.15 | 0.24 |
| Standard error of the variance | 0.87 | 0.13 | 0.85 | 1.90 | 0.01 | 1.79 | 1.46 | 0.11 | 0.81 | 0.27 | 0.15 | 0.39 |

**Table 5.** Descriptive statistics of measured indicators with mean RCA values higher than 0 for each observed code within 2004–2018.

|  | 220860 | | | 220870 | | | 220890 | | |
|---|---|---|---|---|---|---|---|---|---|
|  | **RXA** | **RTA** | **RCA** | **RXA** | **RTA** | **RCA** | **RXA** | **RTA** | **RCA** |
| Nbr. of observations | 15.00 | 15.00 | 15.00 | 15.00 | 15.00 | 15.00 | 15.00 | 15.00 | 15.00 |
| Nbr. of missing values | 0.00 | 0.00 | 0.00 | 0.00 | 0.00 | 0.00 | 0.00 | 0.00 | 0.00 |
| Minimum | −0.40 | −0.65 | −0.68 | 0.04 | −0.09 | −0.06 | 0.34 | −0.98 | −0.34 |
| Maximum | 1.79 | 4.77 | 1.61 | 1.70 | 4.56 | 1.77 | 2.30 | 6.52 | 1.05 |
| First quartile | 0.49 | 0.29 | 0.18 | 0.44 | 0.55 | 0.46 | 1.13 | 0.11 | 0.03 |
| Median | 0.74 | 0.60 | 0.37 | 0.87 | 1.16 | 0.87 | 1.48 | 0.97 | 0.25 |
| Third quartile | 1.12 | 1.85 | 0.89 | 1.09 | 2.00 | 1.15 | 1.77 | 1.84 | 0.53 |
| Mean | 0.76 | 1.28 | 0.51 | 0.84 | 1.55 | 0.81 | 1.44 | 1.33 | 0.27 |
| Variance ($n$) | 0.37 | 2.87 | 0.46 | 0.22 | 1.90 | 0.28 | 0.22 | 3.50 | 0.13 |
| Standard deviation ($n$) | 0.61 | 1.69 | 0.68 | 0.47 | 1.38 | 0.53 | 0.47 | 1.87 | 0.36 |
| Variation coefficient ($n$) | 0.80 | 1.32 | 1.32 | 0.55 | 0.89 | 0.66 | 0.33 | 1.41 | 1.35 |
| Skewness (Pearson) | 0.05 | 1.06 | 0.33 | 0.35 | 1.04 | 0.18 | −0.35 | 1.34 | 0.23 |
| Standard error of the variance | 0.15 | 1.16 | 0.18 | 0.09 | 0.77 | 0.11 | 0.09 | 1.42 | 0.05 |

All remaining product groups experienced negative mean RCA values within the observed time period. We can also clearly see that all of these product groups had higher standard deviations compared to those with positive mean values, explaining their higher overall dynamics. On the other hand, the variation coefficient was negative in all of these product groups. The highest negative mean RCA value was measured for products within category 220820 (spirits obtained by distilling grape wine or grape marc). This outcome can be attributed to the steep decline in its export share (from 19.8% in 2004 to 3.8% in 2018) compared to the decline in its import share (14.8% in 2004 to 9.8% in 2018).

Figure 2 demonstrates how the RCA changed for individual commodities, which clearly reveals a diversified behavior between product groups. However, the most stable values were identified in competitive branches with RCA values above 0 in the last observed year, 2018. The most stable data were observed within category 220870 (liqueurs and cordials). This is the only product group that achieved positive RCA values for almost the whole time period observed. The only exception was the first year of our investigation. On the other hand, the highest volatility was identified in uncompetitive branches of the Slovak spirits industry (as shown in Table 3). The most significant changes were observed within code 220830 (Whiskies).

Based on previous calculations, we conducted separate analyses of "competitive" and "non-competitive" commodities of the Slovak spirits industry in a single market of the EU.

The first group is represented by commodities that had a positive revealed comparative advantage in 2018 (Table 6). Following this approach, two competitive commodities were identified. The highest score was identified for products within code 220870 (Liqueurs and cordials). The second product group with a positive RCA value was code 220860 (Vodka). These two groups represented a 57.2% share of the total Slovak Spirits exports in 2018 (compared to 45.4% in 2004).

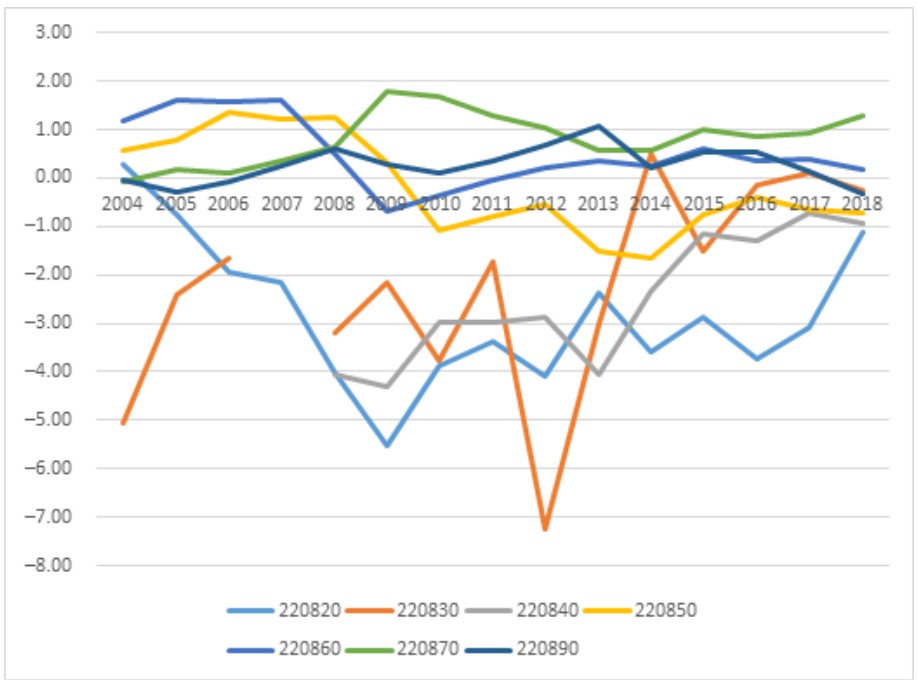

**Figure 2.** Revealed comparative advantage within all investigated branches of the spirits industry in the Slovak Republic from 2004 to 2018.

**Table 6.** Branches of the Slovak spirits industry with an RCA above 0 in 2018.

|  | 220860 | | | 220870 | | |
|---|---|---|---|---|---|---|
|  | **RXA** | **RTA** | **RCA** | **RXA** | **RTA** | **RCA** |
| 2004 | 1.32 | 2.59 | 1.19 | 0.37 | −0.09 | −0.06 |
| 2005 | 1.49 | 3.55 | 1.60 | 0.68 | 0.32 | 0.18 |
| 2006 | 1.79 | 4.77 | 1.59 | 0.04 | 0.11 | 0.11 |
| 2007 | 1.76 | 4.65 | 1.61 | 0.38 | 0.45 | 0.36 |
| 2008 | 0.75 | 0.82 | 0.49 | 0.90 | 1.14 | 0.62 |
| 2009 | −0.40 | −0.65 | −0.68 | 1.70 | 4.56 | 1.77 |
| 2010 | −0.13 | −0.36 | −0.35 | 1.68 | 4.40 | 1.70 |
| 2011 | 0.24 | −0.07 | −0.05 | 1.18 | 2.35 | 1.29 |
| 2012 | 0.74 | 0.40 | 0.21 | 0.87 | 1.53 | 1.02 |
| 2013 | 0.79 | 0.68 | 0.37 | 0.42 | 0.65 | 0.57 |
| 2014 | 0.57 | 0.36 | 0.23 | 0.47 | 0.70 | 0.58 |
| 2015 | 0.92 | 1.11 | 0.59 | 0.97 | 1.65 | 0.99 |
| 2016 | 0.56 | 0.50 | 0.34 | 1.00 | 1.57 | 0.87 |
| 2017 | 0.61 | 0.60 | 0.39 | 0.66 | 1.16 | 0.91 |
| 2018 | 0.43 | 0.22 | 0.16 | 1.31 | 2.67 | 1.28 |

The most important commodity in this group is code 220870 (liqueurs and cordials). Based on previously published export data, import data, and descriptive statistics of the RCA, we identified commodities registered under this code as key elements to the successful competitiveness of the Slovak spirits industry. This product group accounted for 43.8% of the total spirits exports (compared to 27.2% in 2004) and was the only category with a positive RCA value for almost the whole period studied. The only exception was in 2004. The RCA grew steadily between 2004 and 2011. The growth of the export market share of this product group was specific to the time period. Our measurements show the specific behavior for this product group, as its peak competitiveness was observed in 2009, 2010, and 2011. This differs from other branches of the spirits industry. In particular, the economy and international trade were negatively affected by the economic crisis at that time. Food commodities lost their competitive advantage in all EU countries [18],

but products under code 220870 (liqueurs and cordials) behaved in an opposite direction. These products accounted for more than one half of the total Slovak spirits exports in 2009 and 2010 (55.8% in 2009 and 55.9% in 2010).

Code 220860 (Vodka) was the second most competitive group of products in our research in the last observed year, 2018. In contrast to code 220870, code 22060 experienced the worst RCA values in 2009 and 2010. This commodity did not change its position related to international trade within the Slovak spirits industry during the observed years. It occupied the fourth position in exports and fifth position in imports. This branch achieved a revealed comparative advantage above 1 between 2004 and 2008, and it experienced even better outcomes than category 220870 (liqueurs and cordials) during the years mentioned. After two years of decline, it was not able to restore its competitiveness to its previously measured level. According to the historical background of our spirits industry, we considered this category to be one of the commodities that will potentially drive the competitiveness of the Slovak spirits industry in the future. The focus on this commodity is eminent within the Slovak spirits industry, as it accounts for 35.4% of the overall production [36].

The second group of products had negative values of the revealed comparative advantage in 2018 (Table 7). This group is represented by five codes. According to our measurements, most of the branches examined in the Slovak spirits industry are not competitive in a single market. Although these product groups accounted for 71.4% of total Slovak spirits branches, they only represented 42.8% of total exports in 2018.

**Table 7.** Branches of the Slovak spirits industry with a revealed comparative advantage below 0 in 2018.

| | 220820 | | | 220830 | | | 220840 | | | 220850 | | | 220890 | | |
|------|-------|-------|-------|-------|-------|-------|-------|-------|-------|-------|-------|-------|-------|-------|-------|
| | RXA | RTA | RCA | RXA | RTA | RCA | RXA | RTA | RCA | RXA | RTA | RCA | RXA | RTA | RCA |
| 2004 | 0.57 | 0.43 | 0.28 | −6.48 | −0.24 | −5.06 | : | : | : | −0.01 | 0.44 | 0.58 | 1.14 | −0.14 | −0.04 |
| 2005 | −0.49 | −0.70 | −0.76 | −3.91 | −0.20 | −2.39 | : | : | : | −0.12 | 0.48 | 0.79 | 1.12 | −0.98 | −0.28 |
| 2006 | −1.42 | −1.45 | −1.95 | −3.10 | −0.19 | −1.66 | −1.32 | −0.25 | −0.66 | −0.03 | 0.72 | 1.35 | 1.55 | −0.32 | −0.06 |
| 2007 | −1.89 | −1.13 | −2.14 | : | −0.27 | : | : | : | : | 0.07 | 0.76 | 1.22 | 1.70 | 1.23 | 0.25 |
| 2008 | −4.24 | −0.79 | −4.03 | −4.51 | −0.26 | −3.20 | −4.61 | −0.57 | −4.07 | 0.87 | 1.69 | 1.23 | 1.87 | 2.98 | 0.61 |
| 2009 | −4.82 | −2.02 | −5.53 | −3.90 | −0.15 | −2.15 | −5.20 | −0.41 | −4.32 | −0.08 | 0.24 | 0.30 | 1.41 | 0.97 | 0.27 |
| 2010 | −3.80 | −1.05 | −3.87 | −5.19 | −0.24 | −3.77 | −3.73 | −0.45 | −2.97 | −1.20 | −0.60 | −1.09 | 1.48 | 0.41 | 0.10 |
| 2011 | −3.42 | −0.92 | −3.37 | −3.04 | −0.22 | −1.72 | −3.45 | −0.59 | −2.97 | −1.12 | −0.40 | −0.80 | 1.85 | 1.85 | 0.34 |
| 2012 | −4.21 | −0.87 | −4.08 | −8.30 | −0.35 | −7.26 | −3.15 | −0.70 | −2.86 | −1.20 | −0.21 | −0.54 | 2.04 | 3.80 | 0.68 |
| 2013 | −2.39 | −0.90 | −2.38 | −4.18 | −0.30 | −3.02 | −3.64 | −1.49 | −4.06 | −2.05 | −0.46 | −1.52 | 2.30 | 6.52 | 1.05 |
| 2014 | −3.39 | −1.19 | −3.59 | −0.55 | 0.23 | 0.50 | −1.78 | −1.56 | −2.32 | −2.15 | −0.49 | −1.65 | 1.29 | 0.72 | 0.22 |
| 2015 | −2.49 | −1.38 | −2.88 | −2.74 | −0.22 | −1.50 | −0.41 | −1.41 | −1.14 | −1.30 | −0.31 | −0.77 | 1.49 | 1.83 | 0.53 |
| 2016 | −3.03 | −1.95 | −3.72 | −1.35 | −0.04 | −0.14 | −0.54 | −1.54 | −1.30 | −0.99 | −0.19 | −0.40 | 1.10 | 1.25 | 0.53 |
| 2017 | −2.58 | −1.57 | −3.07 | −0.86 | 0.04 | 0.09 | 0.03 | −1.11 | −0.73 | −1.09 | −0.30 | −0.64 | 0.90 | 0.36 | 0.16 |
| 2018 | −0.99 | −0.75 | −1.11 | −1.25 | −0.09 | −0.26 | −0.09 | −1.40 | −0.93 | −1.27 | −0.29 | −0.71 | 0.34 | −0.58 | −0.34 |

Code 220820 (spirits obtained by distilling grape wine or grape marc) was the least competitive branch of the Slovak spirits industry in 2018. This product group was the third most important export commodity within the spirits industry in 2004 (19.8%) but became the second-worst-performing commodity (3.8%) in 2018. A negative mean value was also observed for this commodity, and the standard deviation was the second highest among all observed product groups.

The second-largest negative value of the revealed comparative advantage in 2018 was identified for code 220840 (rum and other spirits obtained by distilling fermented sugarcane products). Commodities identified under this code do not represent common products in the Slovak Republic because there is a shortage of the raw materials required. We did not have complete data available for our investigation in this case. This may also be due to the marginal importance of these commodities. Despite this fact, the export share grew during the investigated years (6.5% in 2018). On the other hand, the import share was higher. This is caused by domestic consumer demand that cannot be saturated by local producers.

Code 220850 (gin and Geneva) achieved the third-highest negative value of the revealed comparative advantage in 2018. Similar to code 220840 (rum and other spirits

obtained by distilling fermented sugarcane products), this commodity is not a commonly manufactured product in the Slovak Republic. Instead, juniper distillate is used to produce the premium "Borovička" distillate, which is made by mixing juniper distillate and neutral alcohol of agricultural origin. This product does not fulfill the criteria for code 220850 because gins are not produced in this way. Gin (distilled gin or London gin) products are made by flavoring or macerating juniper (or other botanicals), according to the regulation 110/2008 of the European Parliament and of the Council. The vast majority of "Borovička" distillate sold in the Slovak Republic does not meet any criteria set by Regulation 110/2008 for juniper distillate because it is made by mixing organoleptically suitable ethyl alcohol of agricultural origin and synthetic aroma.

The third most exported commodity (representing 14.5% of the total Slovak spirits exports) was the product group listed within code 220830 (whiskies). This product group was also not competitive in a single market, according to our measurements for 2018. This category of spirits was one of the most dynamic, as exports rose from 0.1% in 2004 to 14.5% in the last observed year. The import share did not change in the same way, as it rose by only 4% in 14 years. After investigating all of the available data, we can clearly see a change after 2013 (see Figure 2). Investigating the export value numbers, we see a growth of 3.726% between 2013 and 2014. This change may be primarily caused by launching whisky production in one of the distilleries in the country. Roughly 17,000 barrels of whisky [37] are stored within this facility, representing approximately 10% of all Slovak spirits production. Not a single spirit producer within the country would be able to produce whisky in the quantities mentioned. This significantly changed the landscape of whisky production in the Slovak Republic.

The last investigated product group, namely, those under code 220890 (ethyl alcohol of an alcoholic strength of <80% vol, not denatured; spirits and other spirituous beverages), had the highest export share among all investigated categories in 2004. This situation changed during the observed years, and in the last year, the share of overall export was 15.7% (down from 30.4% in 2004). This group represents products not listed in the other categories. It is composed of a rich variety of products that do not create one uniform unit. In this category, the dominance of imports persisted over the whole study period. It is by far the most imported group of products. This may be due to the richness of products made around the world and the actual demand for these products in the local market.

Table 8 provides an answer to our first research question. We investigated the change in competitiveness of individual branches of the Slovak spirits industry in a single market of the EU in 2004 and 2018. The previously mentioned results support this outcome. The change in competitiveness has a negative trend, as there were four competitive branches in 2004 and only two in 2018. Only code 220860 products remained competitive 15 years after the Slovak Republic joined the EU. Code 220860 lost a significant amount of its competitiveness during the observed years, and there is a risk of a complete loss of competitiveness in the future. Code 220870 represents the second-most competitive branch of the Slovak spirits industry in the single market of the EU, according to our calculations. This product group changed its status from uncompetitive to a competitive branch of the Slovak spirits industry within the 15 years analyzed. The competitiveness of this product group was even higher compared to the previously mentioned code 220860 in 2018.

We evaluated the change in total export share of the competitive branches of the Slovak spirits industry in a single market of the EU, and the results are shown in Table 9. This table clearly reveals growth in the export share of the competitive branches of the Slovak spirits industry within the observed territory. This means that, despite stagnation in the total number of competitive branches, the Slovak spirits industry was able to improve its position through the competitive product groups listed under codes 220860 and 228070.

**Table 8.** The change in the number of competitive product groups of the Slovak spirits industry in a single market of the EU.

| Nomenclature Code | RCA | | | | Change |
|---|---|---|---|---|---|
| | 2004 | Competitiveness | 2018 | Competitiveness | |
| 220820 | 0.28 | Competitive | −0.75 | Uncompetitive | Negative |
| 220830 | −5.06 | Uncompetitive | −0.26 | Uncompetitive | x |
| 220840 | 0.66 | Competitive | −0.93 | Uncompetitive | Negative |
| 220850 | 0.58 | Competitive | −0.71 | Uncompetitive | Negative |
| 220860 | 1.19 | Competitive | 0.16 | Competitive | x |
| 220870 | −0.06 | Uncompetitive | 1.28 | Competitive | Positive |
| 220890 | −0.04 | Uncompetitive | −0.34 | Uncompetitive | x |
| Total competitive nomenclature codes | 4 | | 2 | | −2 |

**Table 9.** The change in the total export share of the competitive codes of the Slovak spirits industry in a single market of the EU.

| Year | Share |
|---|---|
| 2004 | 45.4 |
| 2018 | 57.2 |

## 4. Discussion

In our research, we focused on the competitiveness of the Slovak spirits industry in a single market of the EU. Spirits represent a segment of the national economy that has been identified as a "competitive sector" of the Slovak food industry by the "National Concept of the food industry development" [20].

The predisposition of this industry to accelerating the exports of the Slovak Republic is attributable to its nature. Alcoholic beverages were set to achieve global revenues of USD 1,513,925 mil in 2020 [38]. This value is higher than the global revenues of the meat or dairy industry [39,40]. Spirits represent a dominant sector of the alcoholic beverage market in relation to the pure alcohol consumed in the Slovak Republic [41]. Another significant export advantage of spirits is the ratio of the shipping cost to the wholesale price. The higher price of spirits is a benefit, especially in comparison to beer. All of these factors underline the importance of the investigated industry as a key player in the preservation of the Slovak agriculture and food industry.

We can clearly identify a growth in trade value (both imports and exports) within the Slovak spirits industry after the Slovak Republic joined the EU. Both the consumption and production of Slovak spirits is moving toward the European market. Our research revealed rising competitiveness in some commodities over the observed years. The best example is the trade of commodities listed under codes 220830 (whiskies) and 220840 (rum and other spirits obtained by distilling fermented sugarcane products), which are not typical products for the Slovak consumer or the Slovak spirits industry. In the case of whiskies, we see that the industry is adopting a new trend. Despite the current negative RCA value of this commodity, we anticipate that the export value and the overall competitiveness of the whisky sector will grow in the years to come. Commodities listed under code 220840 represent products distilled from sugarcane. Adopting this trend is more difficult given the actual climatic and geographical conditions of the Slovak Republic. On the other hand, we identified a decline in some typical commodities of the Slovak spirits industry. One example is commodities listed under code 220860 (vodka). This group remains one of the key players in trade, but its importance declined during the observed years. This decline in trade is in accordance with the global trend and changes in consumer preferences [42].

In general, there is a clear decline in the total number of competitive commodities produced by companies in the Slovak Republic in a single market. On the other hand,

a positive situation was identified in the total share of competitive product groups in Slovak exports. Despite the decline in the number of competitive categories, the remaining competitive product groups represented 57.2% of the total Slovak spirits exports in 2018 compared to 45.4% in 2004.

According to our research, the Slovak spirits industry is on a trajectory to remain competitive in the single market of the EU. To support the further growth of exports, some measures should be considered. A national strategy focused on the export competitiveness of the Slovak spirits industry is one of the key instruments that policymakers should consider. Following current trends within the industry, it should reflect changes in the tastes of consumers inside and outside the country. The second proposed measure is the adaptation of current legislation to new trends that the industry is facing. National policymakers can influence both European and national legislation. At the European level, it is necessary to take adequate measures to protect traditional regional products. For example, the current regulation that provides the definition, description, presentation, labeling, and protection of the geographical indications of spirit drinks includes 10 traditional Slovak spirits protected by EU law compared to the 86 traditional spirits products registered by France. Next, national legislation should adapt to new challenges that the Slovak spirits industry is facing. Many legislative measures adopted 14 or more years ago restrict the further development of producers, especially smaller ones. One example is the current situation for small spirits producers regulated by Act No. 467/2002 Coll. on the production and marketing of alcohol, as amended. According to this act, small spirits producers have to meet specific conditions. The main benefit is a lower financial obligation to the state authority if the production is limited to 35,000 L of pure alcohol. The most significant disadvantage of this legal regulation is its strict limitations on the products that these small spirits enterprises can produce. In contrast to other EU countries, small spirits producers are not allowed to produce spirits by mixing them with additional products (liquors) and must produce "pure" sprits (vodka). Thus, this legislative regulation prevents small spirits producers from accessing the most lucrative segments of the industry, as our research results show.

There are also many other measures that could positively influence the competitiveness of the Slovak spirits industry, e.g., waste management regulation, carbon footprint, and other environmental impacts of beverage production. These topics are related to the wider definition of the spirits industry, including products made for pharmaceutical purposes, the chemical industry, and transportation. This is beyond the scope of our current study but worth further in-depth research.

The creation of a national strategy with clearly set legislative and supportive mechanisms can create a more competitive and, in the end, more sustainable industry that will also help national agriculture by processing its products of lower quality. The spirits industry represents a strong partner for agriculture and thus also supports the development of rural agricultural areas. As published in the latest research [4], specialization in the agri-food sector may help to improve many macroeconomic indicators. It may also contribute to the sustainable development of the whole agri-food sector. Specialization should be oriented to competitive branches of the agri-food sector. Our research shows that national policies that target the Slovak spirits industry as one of the competitive branches can be justified.

**Author Contributions:** Conceptualization, O.B.; methodology, P.B.; software, O.B.; validation, O.B.; formal analysis, O.B.; investigation, O.B.; resources, O.B.; data curation, I.A. and N.T.; writing—original draft preparation, O.B.; writing—review and editing, N.T.; visualization, O.B.; supervision, P.B.; project administration, I.A. and N.T.; funding acquisition, O.B. All authors have read and agreed to the published version of the manuscript.

**Funding:** This research paper was prepared in the frame of the Erasmus+ Jean Monnet Module project "Economic and Legal Basics of Entrepreneurship in Agrifood Industry" No. 600459-EPP-1-2018-1-SK-EPPJMO-MODULE. With the support of the Erasmus+ Programme of the European Union.

**Institutional Review Board Statement:** Not applicable.

**Informed Consent Statement:** Not applicable.

**Data Availability Statement:** Publicly available datasets were analyzed in this study. The data can be found here: https://appsso.eurostat.ec.europa.eu/nui/show.do?query=BOOKMARK_DS-645593_ QID_-2BFEB81C_UID_-3F171EB0&layout=PERIOD,L,X,0;REPORTER,L,Y,0;PARTNER,C,Z,0;PRO DUCT,L,Z,1;FLOW,L,Z,2;INDICATORS,C,Z,3;&zSelection=DS-645593INDICATORS,VALUE_IN_ EUROS;DS-645593FLOW,1;DS-645593PARTNER,EU27_2020_EXTRA;DS-645593PRODUCT,TOTAL; &rankName1=PARTNER_1_2_-1_2&rankName2=INDICATORS_1_2_-1_2&rankName3=FLOW_1_2 _-1_2&rankName4=PRODUCT_1_2_-1_2&rankName5=PERIOD_1_0_0_0&rankName6=REPORTER _1_2_0_1&sortC=ASC_-1_FIRST&rStp=&cStp=&rDCh=&cDCh=&rDM=true&cDM=true&footnes= false&empty=true&wai=false&time_mode=NONE&time_most_recent=false&lang=EN&cfo=%23%2 3%23%2C%23%23%23.%23%23%23&cxt_bm=1&lang=en and https://www7.statistics.sk/wps/por tal/6285f856-319e-478d-8aef-28f305bcc41f/!ut/p/z1/rVXbdppAFP2V5iGPZM5cgOERUZGAhouQ yEsXIiTUeEkkSe3Xd9SmjRpHVltcS8TZe6-zD3P2oBTdoXSevVb3WV0t5tmjeB6l2tdAd3irhU0AXW2D cx23AzsyMDBAt_sAruoAziCKAZIIhoyhVCwnnr9bvglxFwIAd-DyoRP49IjfSgg4-iDEbhDYuKU340 sAe3ze62Nwup6f-J0hDhM45B8D0tP2LKLJ_Vu__P81XwD2_UexKZZDR7W8FrX1I77di7kAtMC2DQ -DpTfjSwAH_et1wPRVP1SvA2IH7Lh_hwDJ--u3dXn9jEIzvgQg6b8bnOm_67J_4we7_plBEERekoCd kC44FNswiAUk1FGCUpTm83pZP6DRYrzKHpTVVFm-jC9BfD1W0yyvikt4XRX1dP3xH41wteSqpl BsFArT-UThWVEqhJcU1HGeM1xutJd5NUGjRuidF0mxaZMssGyzx3QPgHu2Co7Zi0MjoBRMeq4X Oz6cuExoxpcUmMrlbzf9OuPgnEYq2zBDYHJABO8K3cgnpsFsqx3eCJtDi_DI0wgAPlI4jkypgq3Ja9iG 1j7gk1Q5l6tShe1cH9TgWpYAqMO2mgyo3znTqO1kSvfDUEMjsR_0k4CBCK_XqnhD8XzxPBNnXfRn XDJiYIILTSF6XiqMsFLJ9IIq5VgrND5RszEjqAfoGqXVeHb1ls-u4IpRzoECZkCwYRh0c3iS577Vvxey Wf2gVPNyge6mWZ09Lu6__B7magOsvj09paaIgsW8Lr7X6O49Cy5hVYvTOFe2K_Na5MLzYvIyrdf7 CfGJ7M7_qXlwyWn_jeJizzwmXNcYwYwbhIrfJ8w3Epa1479F43IWf7xmnK6rqlKmYWf3KfsdytLxmv 5oDRRxfzMvLn4CM1SUPg!!/dz/d5/L2dBISEvZ0FBIS9nQSEh/.

**Conflicts of Interest:** The authors declare no conflict of interest.

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
