# Peer review of "Sustainability of the Slovak Spirits Industry in the Single Market of the EU"

_sustainability, doi:10.3390/su13105692_

Round 1
Reviewer 1 Report
The articles focus is on the sustainability of the Slovak spirits industry on EU market. This is achieved by analyzing the competitive advantage of the Slovak producers on the EU single market.
The literature review is well constructed and presents the most important views in the field.
The methodology is suited for the proposed analysis. The data are obtained from reliable sources.
The conclusions are supported by the results.
Author Response
Thank you for the review of the manuscript.
Reviewer 2 Report
Dear Authors,
altogether, I appreciate the manuscript you submitted, but I don't find any reference to sustainability and to the editorial guide lines of Sustainability, but I leave this evaluation to the Editors. The theme of sustainability, just mentioned in the abstract, is not properly developed in the text.
By the way, major weaknesses are:
- RQ(s) is (are) not clearly identified,
- Methodology section should better structured,
- Data exposition may need a table to summarise results.
Minor weaknesses are:
- line 30 justify the sentence,
- line 34 demonstrate the sentence,
- line 40 on which basis you can affirm these study objects,
- line 57 list some of these researchers,
- line 209 this sentence could appear non consistent, meaning is understandable, but it needs an extra explanation,
- line 221 justify the chosen of these measuring methods,
- line 271 a diagram could be helpful to better understand methodology,
- line 273 this sentence could be positioned in the methodology.
Author Response
Thank You for the review of the manuscript and your recommendations.
I would like to summarize the changes made in relation to the recommendations.
Small weaknesses:
- Line 30 justify sentence
References supporting the sentence positioned within the line 31 were added. Our presumption is based on actual articles available at the moment. As for reference, we added two articles:
- (2020). The impact of COVID-19 on food and agriculture in Europe and Central Asia and FAO’s response. FAO. http://www.fao.org/3/ne001en/ne001en.pdf
- S&P Global (2021). Industries Most and Least Impacted by COVID-19 (Probability of Default Perspective) Recovery Insights: March 2021 Update. S&P Global. https://www.spglobal.com/marketintelligence/en/news-insights/blog/industries-most-and-least-impacted-by-covid19-from-a-probability-of-default-perspective
- Line 34 demonstrate the sentence
Further evidence was discussed. Two sentences for explanation are positioned in lines 35-37 : „Food and drink industry employs 4.72 million people and is a leading employer in the EU [1]. This industry is a major employer in fifteen EU countries and the second most important employer in additional seven EU countries [1].”
- Line 40 on which basis you can affirm these study objects
The example of three study objects was put in context by using previous research. We have put two examples of previously published articles for each individual study object. Changes are executed in lines 47-53.
- Line 57 list some of these researchers : Two research articles were listed as a reference for the sentence situated in lines 63-64.
- Line 209 this sentence could appear non consistent, meaning is understandable, but it needs an extra explanation
Further explanation was added as follows (lines 244-253):
“Agricultural products are processed by enterprises situated within the national economy and the main part of the food production is dedicated to the local (national) consumption in developed countries. Developed countries are more oriented on the export of goods and services with higher value added. However, less developed countries are typically oriented on extra-trade of the agri-food products. This is usually caused by the state of the manufacturing sector that is significantly less developed compared to developed countries. Lacing processing capacities, less developed countries are oriented on the export of agricultural products. These agricultural products are often later processed in developed countries in order to justify national consumption. When considering the economic development level of the Central European countries we suggest that the prevailing number of agri-food products is sold on the Single Market of the EU.”
- Line 221 justify the chosen of these measuring methods
Within subchapter 2.4 Research methods the chosen research method was justified by adding sentences referring to the previously conducted research in this area that is also using this approach. Thus, it seems that this method has good potential to describe changes in competitiveness within our object of research, too. Changes are positioned within lines 286-288.
- Line 271 a diagram could be helpful to better understand methodology
The scheme was added to explain the research process, describing individual stages of research. Changes are incorporated to subchapter 2.3 within the manuscript (lines 263-264)
- Line 273 This sentence could be positioned in the methodology
Sentences in lines 272-277 were put to the methodology section (lines 132-137)
Major weaknesses:
- Research questions not clearly identified
We have created a separate subchapter „2.1 Research questions“ consisting of two separate research questions:
- How has changed the number of competitive branches of the Slovak spirits industry on the Single Market of the EU during observed years?
- How has the total export share of competitive branches of the Slovak spirits industry changed during observed years concerning the Single Market of the EU?
Both research questions are based on the main aim of the manuscript. Each question contains a short description and its importance toward the competitiveness of the Slovak spirits industry. Changes are made within lines 139-155.
- Methodology section should better structured
The chapter dedicated to methodology and methods was changed in formal and content in order to improve the overall readability and quality of the dedicated chapter. We have created subchapters of the mentioned chapter. These changes are in line with previous change regarding research questions. The following subchapters were created:
- Research questions
- Research object
- Research methodology
- Research methods
Sentences positioned within lines 254-270 were moved into the new subchapter „2.3 Research methodology“. The new position is within lines 267-283.
Some other minor formal changes were implemented in order to improve the quality of the manuscript, i.e. italic style was added to the nomenclature explanations within lines 188-219.
- Data exposition may need a table to summarise results
Results were summarized in a more complex and visual way by adding two separate tables. Table 8. represents summarized research outcomes of the research devoted to the first research question (number of competitive branches of the Slovak spirits industry). A short description of the table was added, too. The second table (Table 9.) represents outcomes of the research-oriented on the second research question, as stated in subchapter 2.1. We added also a short description of the table. Changes are located within lines 574 and 600.
- Missing connection to sustainability
To support the connection of the sustainability of the food industry and the competitiveness of individual branches we have added additional text into the introduction section (37-42) and discussion (674-679). We added also a reference to the latest research dealing with the competitiveness of individual branches of the food industry and its impact on the macroeconomic indicators and sustainability of the food industry within national economies.
Round 2
Reviewer 2 Report
I consider the changes you bring to your article enough.
I wish you the best with this article